# A High Performance and Robust FPGA Implementation of a Driver State Monitoring Application

**DOI:** 10.3390/s23146344

**Published:** 2023-07-12

**Authors:** P. Christakos, N. Petrellis, P. Mousouliotis, G. Keramidas, C. P. Antonopoulos, N. Voros

**Affiliations:** 1Electrical and Computer Engineering, University of Peloponnese, 263 34 Patras, Greece; p.christakos@esdalab.ece.uop.gr (P.C.); ch.antonop@uop.gr (C.P.A.); voros@uop.gr (N.V.); 2Electrical and Computer Engineering, Aristotle University of Thessaloniki, 541 24 Thessaloniki, Greece; p.mousouliotis@esdalab.ece.uop.gr; 3Computer Science, Aristotle University of Thessaloniki, 541 24 Thessaloniki, Greece; gkeramidas@csd.auth.gr

**Keywords:** shape alignment, driver drowsiness, face detection validity, machine learning, hardware acceleration, Vitis HLS

## Abstract

A high-performance Driver State Monitoring (DSM) application for the detection of driver drowsiness is presented in this paper. The popular Ensemble of Regression Trees (ERTs) machine learning method has been employed for the alignment of 68 facial landmarks. Open-source implementation of ERTs for facial shape alignment has been ported to different platforms and adapted for the acceleration of the frame processing speed using reconfigurable hardware. Reducing the frame processing latency saves time that can be used to apply frame-to-frame facial shape coherency rules. False face detection and false shape estimations can be ignored for higher robustness and accuracy in the operation of the DSM application without sacrificing the frame processing rate that can reach 65 frames per second. The sensitivity and precision in yawning recognition can reach 93% and 97%, respectively. The implementation of the employed DSM algorithm in reconfigurable hardware is challenging since the kernel arguments require large data transfers and the degree of data reuse in the computational kernel is low. Hence, unconventional hardware acceleration techniques have been employed that can also be useful for the acceleration of several other machine learning applications that require large data transfers to their kernels with low reusability.

## 1. Introduction

Face landmark alignment is an operation exploited in several applications ([1,2,3,4]). More specifically, an interesting approach is presented in [3] where the authors use facial shape alignment to capture expressions that can predict strokes. Additionally, in [4], Pascual et al. implement an application to recognize facial emotions. Although there have been numerous Machine Learning (ML)- and Artificial Intelligence (AI)-based approaches for the implementation of face or pose estimation on Central Processing Units (CPUs) or Graphics Processing Units (GPUs) platforms, the approaches that are based on reconfigurable hardware are limited. Field Programmable Gate Arrays (FPGAs) can be exploited to implement several computationally intensive tasks in hardware. For example, one of the authors of this paper has implemented a Convolutional Neural Network (CNN) grinder in FPGA as described in [5]. Multiply-Accumulate operations required, e.g., in digital filters have also been proposed for FPGA implementation [6]. In [7], Kosuge et al. implement a pose estimation algorithm on a Xilinx FPGA platform reporting a power consumption equal to 6 W. The authors in [7] developed an application for pose estimation using a Xilinx ZYBO7010 FPGA where inputs from different sensors are combined. The authors of [8] implement a face recognition algorithm using a Xilinx platform and achieve a processing speed of 45 frames-per-second (fps). In [9], an algorithm is presented that can be executed on an embedded platform (Xilinx FPGA based on ARM A9 processor) that estimates the pose of the hand using 23 landmark points reporting a 30 fps rate. In [10], J. Goenetxea et al. developed a 3D face model tracking application using 68 landmarks achieving a rate of approximately 30 fps. The application presented in [10] is tested on several commercial devices such as iPhones. A pose estimation system for Unmanned Aerial Vehicles (UAVs) is implemented in [11]. It is based on the use of 16 landmarks and is implemented on a Zynq-7000 board. In [12], Z. Ling et al. developed a system to estimate the position of the head, achieving a rate of 16 fps with 68 landmarks and their experiments run on an Altera Cyclone V FPGA. In [13], an FPGA-based pedestrian detection approach under strong distortions is presented.

In this paper, the proposed Driver State Monitoring (DSM) system detects driver drowsiness by recognizing yawning from the distance of specific landmarks around the mouth. The employed facial shape alignment method is based on Kazemi and Sullivan approach [14]. In this approach, an Ensemble of Regression Trees (ERT) machine learning model has been defined to align 68 facial landmarks in 1 ms (as advertised by the authors). The faces can have a maximum tilt of about ±20°. The high speed achieved by the ERT method, as well as the fact that it has been employed in various popular image processing libraries such as DLIB and Deformable Shape Tracking (DEST) [15,16], was the reason we also based the implementation of our DSM system on ERTs. Moreover, open-source code for the implementation of ERTs was also available [15].

Numerous other approaches for different applications such as face expression and emotion recognition are based on the ERT model presented in [14]. While the approach presented in [14] concerns 2D face shape alignment, X. Jin in [17] extended this method in [17] to detect 68 landmarks on face images based on a 3D model. The illumination of the images is also taken into consideration in [17]. The authors in [18] were also based on [14] to correct the face shapes according to the estimated landmarks positions. In [19], the authors used the ERT model to recognize the area around the eyes for measuring the body temperature, in order to avoid congestion, and to protect from COVID-19 virus spread. Islam and Baek [20] exploited the approach presented in [14] to detect the biological human age from faces with occlusion. In [21], the authors used the ERT model to detect face expressions by calculating the geometric center of each eye and the distance between them. Drowsiness indicated from the head position, eye features, and facial expressions is detected by Pandey and Muppalaneni in [22] based on the approach of [14]. The authors attempted to detect drowsiness as early as possible to reduce crash accidents. In [23], the Kazemi and Sullivan approach [14] is employed to detect facial physiological information from face shapes in videos frames, depending on the features of eyes, nose, mouth, and face contour. The approach of Sergio Pulido-Castro et al. [24] is also based on [14] and its aim is to align the landmarks on a face and recognize facial expressions and emotions. The acceleration and robustness techniques presented in this paper for the original 2D shape alignment method with its limitations (no support for 3D models, occlusion and tilt higher than ±20°) can be exploited with no modifications in the approaches referenced above.

In recent driver drowsiness detection approaches, various AI or ML techniques have been employed as reported in the [25] survey paper: OpenCV library services, Viola-Jones algorithm [26], Convolutional Neural Networks (CNNs), Haar Cascade Classifier [27], Support Vector Machines (SVMs) [28], etc. The yawning detection classification accuracy reported in this review paper ranges from 75% to 92.5%. In [29], modern drowsiness detection algorithms based on computer vision are compared. Half of the compared approaches in [29] concern the detection of sleepy eye blinks and the rest of them are based on yawning or both. The approaches for eye blink detection show an accuracy between 81% and 97.8%, while the classification accuracy of the mouth-based detections is not reported.

Some driver drowsiness detection algorithms have been based on embedded platforms. For example, in [30], a smart camera embedded platform, called APEX from CogniVue Corporation is employed to detect yawning, reporting a 3 fps rate. Their approach is based on a modified implementation of the Viola–Jones algorithm for face and mouth detections with a yawning detection rate between 65% and 75%. In [31], driver drowsiness detection based on CNNs is presented and implemented on a Raspberry PI platform with an accuracy between 80% and 98%. The achieved frame rate is not reported. In a recent approach presented in [32], the instruction set of a RISCV based System on Chip (SoC) is extended for driver drowsiness detection. A frame rate of 4.33 fps is reached with an improvement of 1.7× compared to their implementation on a baseline processor (2.56 fps). The accuracy is 63% and 81% on the validation data used during training, respectively. In the seminal work presented by Kowalski et al., in [33]. the proposed Deep Alignment Network (DAN) is actually a CNN for the alignment of 68 facial landmarks with a very small error (less than 5% according to [1]).

The referenced DSM systems are either based on low speed architectures such as CNNs or machine learning techniques for shape alignment that are not optimized for speed, e.g., they are implemented in software or on GPUs in their higher speed versions. Moreover, none of the referenced approaches consider shape alignment in nighttime conditions. Focus on the face displayed in a frame can easily be lost if, for example, the system depends on a face recognition service like the one offered by OpenCV. Our approach concerns the development of a DSM system that overcomes these issues and proposes hardware acceleration and robust operation solutions that can also be exploited by other applications based on ERTs. Although our approach can detect yawning, distraction, and sleepy eye blinks, this work focuses on the successful detection of yawning. It is based on the ERT model defined in [14] and its Deformable Shape Tracking (DEST) implementation [15,16]. Our aim was to port the ERT model presented in [14] to a reconfigurable (FPGA-based) hardware platform in order to accelerate computationally intensive operations. The latency reduction in facial shape alignment allows the incorporation of high-level rules that can recognize failures in face detection and shape alignment in a frame-by-frame basis and rejects the corresponding frames. In this way, a robust operation with acceptable accuracy and high frame processing rate is achieved.

The original DEST video processing application that estimates the facial landmarks in video stream frames was ported to the target FPGA platform (Xilinx ZynqMP Ultrascale+) following a number of successive transformations: (a) the video processing application was profiled in order to calculate the computationally intensive operations and the bottlenecks of the system, (b) based on the profiling results, the code was reconstructed in Ubuntu environment to support an efficient hardware acceleration of the system (the face landmark prediction routine and the nested calls have been flattened, the Eigen C++ library calls have been replaced with high-performing C code), and (c) the resulting code structure was easily ported to Xilinx Vitis environment for hardware implementation through the High Level Synthesis (HLS) tool (the GNU C++ and Vitis compilers are compatible with the ones used in the Ubuntu version). The acceleration achieved with the replacement of Eigen library calls alone (without any hardware support) is significant since the latency *L_f_* of the face landmark prediction routine was reduced from 116 ms to less than 0.5 ms on an Intel i5 processor as described in our prior work [34]. However, adding face recognition validation rules at the top level, the latency of the software implementation increased to approximately 40 ms on an ARM A53 processor core of the target Xilinx ZCU102 board. When hardware acceleration techniques were applied to the target board, the latency was reduced to 16 ms (>62 fps). This was accomplished with the aid of hardware acceleration techniques that were initially specified in [35] and their extension is discussed in this paper. The specific face alignment algorithm that was implemented in hardware is not a typical hardware acceleration problem since the hardware kernel accepts as arguments large-sized buffers and does not perform many repetitive operations on these data. Therefore, significant effort is given to the development of an acceleration technique that would reduce the latency of bulk data transfers from the software (ARM processor) to the hardware kernel (FPGA).

The key contributions of our paper can be summarized as follows: (a) hardware and software acceleration techniques have been applied to open-source code available for ERT face alignment applications, (b) the significant frame processing latency reduction allowed the application of face alignment validation rules that increased the robustness and the accuracy of the system, (c) high precision and sensitivity are achieved without sacrificing the speed, which can reach 65 fps, (d) the unconventional hardware acceleration techniques that have been employed can also be used in the acceleration of other machine learning applications that require the transfer of a large volume of data to their hardware kernels, and (e) a new dataset has been developed to train and test facial shape alignment models in nighttime conditions.

This work differs from our previous works in the following: (a) the software accelerated ERT implementation described in [34] was ported to Xilinx Vitis HLS and implemented as a hardware accelerated application, (b) the hardware acceleration techniques proposed in [35] were implemented and tested leading to an optimal acceleration scheme that reduces frame processing latency by 60%, (c) rules that increase the robustness of the system were added to reject the frames where shape alignment failed, increasing the robustness and accuracy, and (d) the FPGA resources, accuracy, speed, and power consumption have been experimentally measured.

This paper is structured as follows. In Section 2, the experimental setup and datasets used are described. Then, the ERT background and the proposed methods are described in Section 3. More specifically, a brief introduction of the ERT model is presented in Section 3.1, and the system functions and architecture are described in Section 3.2. The employed hardware acceleration techniques are discussed in Section 3.3. Then, the frame-to-frame coherency rules that are applied at the top level of this application for higher robustness and accuracy are described in Section 3.4. The experimental results are presented in Section 4 and discussion of these results follows in Section 5. The conclusions can be found in Section 6.

## 2. Experimental Setup and Datasets

The algorithms described in this paper have been evaluated on two target platforms: an Ubuntu-based Intel i7 environment and on a Xilinx ZCU102 platform (ZynqMP Ultrascale+ with XCZU9EG-2FFVB1156E FPGA) with the programmable logic operating in the hardware kernel with a frequency clock of 150 MHz. The system was developed in C/C++ in the framework of Xilinx Vitis HLS 2020.1 design tool. Petalinux 2020.1 operating system has been installed on one of the Xilinx ZCU102 board available in our lab at Patras, Greece. A variety of appropriate codecs and libraries (such as ffmpeg) were included when building the Petalinux image to support the functionality of the developed system.

Videos from the dataset YawDD [36] have been initially employed to test the developed methods. This dataset contains 360 videos captured within a car environment with drivers (of both genders and various races) pretending to drive and yawn. These videos have a frame rate of 30 fps with AVI format and a resolution of 640 (width) × 480 (height). In some of the videos, the camera is mounted on the internal mirror, while in others it has been placed on the dash; thus, the driver’s face is captured from different angles. The drawback of the YawDD dataset is that the videos are all captured in daytime and within stationary cars. To overcome these non-realistic conditions, we have developed an additional dataset that is made public, called NighTime, Yawning, Microsleep, Eyeblink Detection (NITYMED) [37] available from IEEE data port as well as other repositories. NITYMED offers 130 videos with 20 drivers of different genders and face features (glasses, hair color, etc.). All the videos in NITYMED were captured in real cars moving in night-time, since driver drowsiness is expected to occur more often at that time of the day. The videos in NITYMED are offered in two mp4 formats: HDTV720 with a resolution of 1280 × 720 and HD resolution (1920 × 1080). All videos have a frame rate of 25 fps.

## 3. ERT Background and Proposed Methods

### 3.1. Background of the ERT Model

The face alignment method presented in [14] is based on the ERT model. In this method, the mean position of the landmarks stored in the trained model is gradually corrected in a number of cascade stages. In each cascade stage, a sparse representation of the input frame consisting of a few hundreds of reference pixels is used and a number of regression trees is successively visited. In each regression tree node, the gray level intensity of a pair of reference pixels is compared to decide the next node of the tree that has to be visited. When a leaf is reached, the corresponding correction factor is loaded from the trained model (different in each leaf of a tree) and this factor is used to correct the current landmark positions. In [14], the focus was given on the training of the regression trees while in this paper we focus on the application of a trained model for the face shape alignment in successive video frames in runtime. A more formal definition of the training of the ERT model follows. Let *p* be the number of face landmarks at a frame *frm*, and *x_i_* be a Cartesian coordinates pair. The shape $S\in R^{2p}$, drawn by these landmarks is defined as:(1)$$S=\left\{x_{0},x_{1},..,x_{p}\right\}$$

The training begins by initializing the shape *S* to default positions. The exact position of the landmarks in the frame, which is currently examined, is refined by a sequence of *T_cs_* cascaded regressors. The current shape estimate in the regressor *t* is denoted as ${\hat{S}}^{\left(t\right)}$ (*t* = 1,…, *T_cs_*). To continue with the next regressor, it is needed to estimate a correction factor *r_t_* that depends on the frame *frm* and the current estimate ${\hat{S}}^{\left(t\right)}$. This correction factor *r_t_* is added to ${\hat{S}}^{\left(t\right)}$ in order to update the shape of the face in the next regressor *t* + 1: ${\hat{S}}^{\left(t+1\right)}$. The *r_t_* value is calculated based on the intensities of reference pixels that are defined relative to the current shape ${\hat{S}}^{\left(t\right)}$. Every correction factor *r_t_* is updated using a gradient tree boosting algorithm by adding the sum of square error loss. This training algorithm works with the group $\left(I_{\pi_{i}},\;{\hat{S}}_{i}^{\left(t\right)},{\Delta S}_{i}^{\left(t\right)}\right)$. The training data is a set of *N* images: $I_{\pi_{\iota }},\;0\leq \pi_{i}<N.$ ${\hat{S}}_{i}^{\left(t\right)}$ is the shape of any training frame with *i*$\;\neq \pi_{i}$. The residual ${\Delta S}_{i}^{\left(t+1\right)}$ in the regressor *r_t+_*_1_ is estimated as [14]:(2)$${\Delta S}_{i}^{\left(t+1\right)}=S_{\pi_{i}}-{\hat{S}}_{i}^{\left(t+1\right)}$$

The residuals are exploited by the gradient of the loss function and are calculated for all training samples. The shape estimation for the next regressor stage is performed as [14]:(3)$${\hat{S}}^{\left(t+1\right)}={\hat{S}}^{\left(t\right)}+r_{t}\left(I_{\pi_{i}},{\hat{S}}_{i}^{\left(t\right)}\right)$$

Then, for *K* tests (*k* = 1, …, *K*) in a regression tree and for *N* training frames (*i* = 1,…, *N*) in each test, the following formula is iterated:(4)$$r_{ik}={\Delta S}_{i}^{\left(t\right)}-f_{k-1}\left(I_{\pi_{i}},{\hat{S}}_{i}^{\left(t\right)}\right)$$

In the *k*-th iteration, a regression tree is fitted to *r_ik_* using a weak regression function *g_k_*, thus *f_k_* is updated as follows [14]:(5)$$f_{k}\left(I,{\hat{S}}^{\left(t\right)}\right)=f_{k-1}\left(I,{\hat{S}}^{\left(t\right)}\right)+lr\cdot g_{k}\left(I,{\hat{S}}^{\left(t\right)}\right)$$
where *lr <* 1 is the learning rate used to avoid overfitting. The final correction factor *r_t_* of the t-regressor is equal to *f_K_*.

In runtime, the algorithm begins by initializing the shape *S* to the mean landmark positions mined from the trained model. The exact position of the landmarks in the frame which is currently examined, is again refined by a sequence of *T_cs_* cascaded regressors similarly to the training procedure.

The next node that should be selected in a regression tree of a specific cascade stage is estimated by comparing the difference in the gray level intensity between two reference pixels *p*_1_ and *p*_2_ with a threshold *T_h_*. Different threshold *T_h_* values are stored for every regression tree node in the trained model. The image is warped to match the mean shape of the face within a process called Similarity Transform (ST). If *q* is a frame pixel and its neighboring landmark has index *k_q_*, their distance *δx_q_* is represented as $\delta x_{q}=\left\|q-x_{k_{q}}\right\|.\;$The pixel *q′* in the original frame *frm* that corresponds to *q* in the mean shape is estimated as [14]:(6)$$q^{\prime }=x_{i,k_{q}}+\frac{1}{s_{i}}R_{i}^{T}\delta x_{q}$$

The *s_i_* and *R_i_* are scale and rotation factors, respectively, used in the ST to warp the shape and fit the face. The minimization of the mean square error, between the actual *q′* value and the one estimated with Equation (6), is used to calculate the optimal values for *s_i_* and *R_i_*.

### 3.2. System Architecture and Operation

As noted, the DSM module presented in this paper is based on the DEST video tracking application [15] that matches landmarks on faces that are detected in frames from video or camera streams. Specifically, 68 facial landmarks are aligned as shown in Figure 1. The Mouth Aspect Ratio (*MAR*) of the vertical (*M_v_*) and the horizontal (*M_h_*) distance of specific mouth landmarks can be used to detect yawning if *MAR* is higher than a threshold *T_M_* for a number of successive frames:(7)$$MAR=\frac{M_{v}}{M_{h}}>T_{M}$$

Similarly, closed or sleepy eyes can be recognized if the average Eye Aspect Ratio (*EAR*) of the eyes’ vertical (*E_v_*_1_, *E_v_*_2_) and horizontal (*E_h_*_1_, *E_h_*_2_) distances is lower than a threshold *T_E_* for a period of time:(8)$$EAR=\frac{1}{2}\left(\frac{E_{v1}}{E_{h1}}+\frac{E_{v2}}{E_{h2}}\right)<T_{E}$$

The parameter PERCLOS is the percentage of time that the eyes are closed and it is used to discriminate between a normal fast eye blink and an extended sleepy eye closure. Similarly, the percentage of time that MAR is higher than a threshold *T_M_* is used to distinguish a yawning from speech. In speech, the mouth is open and closed for shorter periods. For higher accuracy, more sophisticated ML techniques can also be employed to detect yawning or sleepy eye blinks instead of simply comparing with thresholds. 

The dimensions of the face shape and the loss of focus can also be used to detect microsleeps or driver distraction. More specifically, during microsleeps the driver’s head is slowly leaning forward and then suddenly, the driver wakes up, raising his head. Therefore, a microsleep can be recognized if the face shape height *H_f_* (Figure 1) is gradually decreased with a potential loss of focus (since no face is recognized by OpenCV when the head is down). Then, the face suddenly reappears to its previous normal position. Driver distraction can be recognized when the driver continuously talks or turns his head right and left without focusing ahead. Again, the shape dimensions, mouth pattern sequences, and loss of focus can be used to detect this situation. In this work, we focus on yawning detection.

The video stream is defined during initialization and the frames are analyzed as follows. If face landmark alignment is required in a specific video frame, face detection takes place using the OpenCV library. For higher processing speed, a new face detection may be avoided in a few subsequent frames. In this case, the face is presumed to exist in an extended bounding box around the estimated landmarks from the previous frame. Landmark alignment is then applied in the bounding box returned by OpenCV using the DEST hierarchy of nested functions called: *predict*(). The Similarity Transform (ST) process, described in Section 3.1, has to be practiced on the detected face bounding box to match its coordinates to those of the mean shape stored in the trained model. This model includes a set of regression trees in every cascade stage and the parameter values of each tree node are available from the training procedure.

As mentioned, the implementation of the ERT method [14] in the open-source DEST package [15] was reconstructed to support acceleration of the computational intensive operations with reconfigurable hardware. The DEST implementation depends on the Eigen library that simplifies complicated matrix operations. The use of Eigen library calls in the original DEST source code protects from overflows and rounding errors, simplifies type conversions, etc. Eigen operators are overloaded to support various numeric types such as complex numbers, vectors, matrices, etc. The computational cost of these facilities is a reduction in the system performance. Furthermore, the Eigen classes are not suitable for hardware synthesis. To address this problem, the source code was simplified (re-written) replacing Eigen classes with ANSI C types that can be successfully ported to hardware using the Xilinx Vitis environment [34]. Moreover, loops and large data copies were also optimized as will be explained in the next section. 

To accelerate the DEST video tracking application in an efficient way, it was mandatory to profile the latency of the system components. To achieve this goal, the system was decomposed in potentially overlapping subsystems, estimating the bottlenecks and any resource limitations that occur if such a subsystem is implemented in hardware. The latency was profiled measuring the time from the beginning to the end of each subsystem operation. In this way, the computational intensive sub-functions were identified. Another factor that slows down the DEST video tracking application is the input/output argument transfer between the host and the subsystems implemented in hardware kernels. This is due to the limited FPGA Block Random Access Memory (BRAM) resources and the Advanced eXtensible Interface (AXI) bus port width. One major target of this work was to optimize big data transfer from the ARM cores to BRAMs with the lowest latency. 

The profiling analysis indicated that the most computational intensive operation in the original DEST video tracking application is the *predict*() function which incorporates three nested routines as shown in Figure 2: *Tracker*::*predict*(), *Regressor*::*predict*() and *Tree*::*predict*(). *Tracker*::*predict*() is the top level function. Its input is the image frame *frm* and the coordinates of the bounding box of the recognized face by OpenCV. This routine returns the final face landmark coordinates. The current shape estimate *S* is initialized to the mean shape (that is stored in the trained model), at the beginning of the *Tracker*::*predict*() routine. Then, the *Regressor*::*predict*() is called inside *Tracker*::*predict*(), once for every cascade stage. In our implementation, the default number of cascade stages is *T_cs_* = 10. The *Regressor*::*predict*() accepts as input the image *frm*, the face bounding box coordinates estimated by OpenCV and the current approximation of the shape estimate *S* after applying the correction factor *f* (see Equation (5)). 

The *Regressor*::*predict*() calculates the residuals *s_r_*, that are used to optimize the estimate *f* in the *Tracker*::*predict*(). The *Regressor*::*predict*() routine employes ST to match the current landmark shape to the mean coordinates of the trained model. Then, the gray level intensities of a number of reference pixels are read from a sparse representation of the *frm* frame. The *Tree*::*predict*() function is invoked for all the stored *N_tr_* binary regressor trees. In this implementation, the default number of regressor tress is *N_tr_* = 500. Every *Tree*::*predict*() computes a correction factor vector called mean residuals *m_r_*. A different *m_r_* vector is stored in the leaves of each regression tree. Each tree is traversed from the root to the leaves in the *Tree*::*predict*() routine, following a path defined dynamically: at each tree node, the difference in the gray level intensities of a pair of reference pixels that are indexed in the trained model is compared to a threshold *T_h_* as already described in Section 3.1. Either the right or left direction of the current node in the binary tree is selected, depending on the threshold *T_h_* that is also stored in the trained model. Each tree has depth equal to *T_d_* = 5 and thus, $2^{T_{d}}-1=31$ nodes.

In the approach presented in this paper, a new *Predict_kernel*() routine was developed including operations from the *Regressor*::*predict* (and the nested *Tree*::*predict*). The *Predict_kernel*() routine was implemented using ANSI C commands and a code structure that can be easily consumed by the HLS tool. 

All the parameters of the trained model that were continuously accessed throughout the operation of the original DEST application are now loaded during initialization. All the model parameters are stored into buffers within the new *predict_prepare*() routine that is called when the system is powered up. More specifically, the number of the *N_tr_* regression trees and their node values are read: the threshold *T_h_*, the mean residual *m_r_*, and the indices of the reference pixels that their intensity has to be compared with *T_h_*. Moreover, the *LM* = 68 coordinates of the mean shape landmarks and the *N_c_* reference pixel coordinates of the sparse image are also read from the trained model within *predict_prepare*(). 

The parameters that are not read once but have to be read again each time the *Tracker*::*predict*() is called are the default mean shape estimate and the mean residuals. These values have to be re-initialized since their value is modified during the processing of each frame. Pointers to these arguments as well as to frame-specific information (image buffer, image dimensions, position of the detected face) are passed to the *Predict_kernel*() routine. The final landmark shape is returned by *Predict_kernel*() and its homogeneous coordinates are converted to the absolute coordinates in order to display the landmarks on the image frame under test.

The whole *Regressor*::*predict*() function was initially considered as a candidate for hardware implementation but it was soon abandoned. The main reason for not porting the whole *Regressor*::*predict*() routine in hardware is that it contains functions that do not reuse data and require a large number of resources. More specifically, it includes ST and the function that reads the reference pixel intensities. These two functions are quite complicated and thus, require a large number of resources. Moreover, their latency is relatively short since they do not have repetitive calculations. Finally, *Regressor*::*predict*() requires a large number (more than 20) of big argument buffers that would not fit to BRAM. Forcing the hardware kernel to access the values of these buffers from the common RAM that is accessed both from the processing and the Programmable Logic (PL) units of the FPGA, would cause a significant bottleneck.

For these reasons, *Predict_kernel*() was defined as a candidate for hardware implementation with a subset of operations from the *Regressor*::*predict*(). The *Predict_kernel*() routine consists from the loop that visits the regression trees and it is called once for every cascade stage. The ST and the routine that reads the reference pixel intensities are left out of this routine (see Figure 2). Its arguments are only the parameters that are stored in each regression tree node (five buffers). The specific steps of the *Predict_kernel*() routine are presented in Algorithm 1. The buffer arguments are listed in line 1: the residuals *s_r_*, the mean residuals (correction factors) *m_r_*, the intensities of the reference pixels that will be compared in each node with the corresponding threshold *T_h_*, and the indices to the next regression tree node that has to be visited according to the results of the comparison with *T_h_*. These indices of the tree nodes are stored in two vectors called *split*1 and *split*2. 

The buffers that are transferred to BRAMs and their specific sizes in the default ERT model (M0) are: (a) the reference pixel intensities (size: *N_c_* floating point numbers, e.g., 600 × 32 bits), (b) the *split*1 and *split*2 buffers (*N_tr_* trees × (2*^Td^* − 1) nodes × short integer size, e.g., 500 × 31 × 16= 15,552 × 16 bits for each one of *split*1 and *split*2), and (c) the *T_h_* buffer (*N_tr_* trees × (2*^Td^* − 1) nodes × floating point size, e.g., 500 × 31 × 32 bits). The mean residuals’ buffer (*m_r_*) is too large (*N_tr_* trees × (2*^Td^* − 1) nodes × 136 landmark coordinates × floating point size, e.g., 500 × 31 × 136 × 32 = 15,552 × 32 bits) to fit in a BRAM; therefore, it has to be accessed from the common DRAM. The BRAMs are created and the corresponding arguments that will be stored there are copied through a single or double, wide data port as will be explained in the next section, using optimized loops in steps 2 to 6 of Algorithm 1. Each argument is considered to consist of two data segments: the first data segment stores parameters from the first *N_tr_*/2 trees and the second data segment from the last *N_tr_*/2 trees. If double data ports are used for each argument, the two data segments of each argument are processed in two parallel loops in step 7. If a single data port is employed for each argument, the two loops of step 7 will simply not be executed in parallel. The partial correction factors *s_r_*_1_, *s_r_*_2_ are then added to the previous value of the overall correction factor *s_r_*, which is returned to the caller of the *Predict_kernel*() routine.
**Algorithm 1** Predict_kernel() algorithm implemented as a hardware kernel.**Predict_kernel**()1.Read arguments from the host (s_r_, split1, split2, T_h_, m_r_, reference pixel intensities)2.Create BRAM arrays for split1, split2, T_h_, and reference pixel intensities3.Loop to store pixel intensities4.Loop to store split1 (from a single or double, wide data port)5.Loop to store split2 (from a single or double, wide data port)6.Loop to store T_h_ thresholds (from a single or double, wide data port)7.Parallelize the estimation of the correction factor to take advantage of double data ports:a.For every tree, update the correction factor s_r1_ (1st data segment)b.For every tree, update the correction factor s_r2_ (2nd data segment)
8.Update the correction factor s_r_ = s_r_ + (s_r1_ + s_r2_)5.Return s_r_ to the host

### 3.3. Hardware Acceleration Methods

The employed hardware acceleration techniques are listed in Table 1. Based on the system component profiling, the hardware implementation of initially the *Regressor*::*predict*() function and then of the *Predict_kernel*() was carried out using Xilinx Vitis High Level Synthesis (HLS) tool. 

The most important optimization concerned the efficient data buffering in order to reduce data transfer latency by more than 20 ms from ARM core to the hardware kernel. The widest AXI bus data port that can be defined is exploited to read concurrently the maximum range of data instead of only a single element during the argument transfer. Xilinx supports the technology to feed any time the kernel with *N_b_* bits of the argument. More specifically, if an argument element requires an m-bit representation, it is feasible to transfer *N_b_*/*m* elements in every clock cycle. The acceleration that can be achieved using this technique can be estimated as follows. If the latency of accessing the main memory for reading a single element of m-bits is *t_m_*, then the latency *L_m_* for accessing N_b_/m elements is:(9)$$L_{m}=\frac{N_{b}t_{m}}{m}$$

Exploiting wide ports that can transfer *N_b_* bits in one clock cycle in a local BRAM and assuming that the BRAM access time for reading a single element of m-bits is *t_bm_*, the latency *L_bm_* for reading the N_b_/m elements from BRAM is:(10)$$L_{bm}=t_{m}+\frac{N_{b}t_{bm}}{m}$$

It is expected that the second term in Equation (10) is much smaller than *L_m_* since *t_bm_* << *t_m_*. Therefore, the hardware acceleration (HA1) achieved in argument passing is:(11)$$HA1=\frac{L_{m}}{L_{bm}}$$

For example, if bytes are transferred, then the value of *m* is 8, while if integers or single precision floating point numbers are transferred, then *m* = 32. To store these data in parallel, it is important to declare an array partition with multiple ports. This implementation requires a data type with arbitrary precision. The Xilinx *ap_int<N>* type provides the flexibility to slice the argument buffers into equal data pieces depending on the application requirements. For example, if it is assumed that 32 elements of 64 bits need to be stored in parallel, the kernel argument can be defined as *ap_int<1024>* and 16 elements can be read at once. Two clock cycles are only needed to read all the 32 elements. The next step is to declare an array of 64-bit elements with dimensions 16 × 2, which will be partitioned using the OpenCL ARRAY_PARTITION pragma in order to achieve maximum parallelization. With this technique any data conflict is avoided, and the data are read with the minimum possible latency. 

The OpenCL array partition operation can be optimized depending on the application. The type of the partition, the dimensions of the array that will be partitioned (all the array or a part of it), and the number of partitions that will be created can be defined with OpenCL. In this approach, the most efficient way to partition the BRAM arrays was in block mode in order to achieve the maximum parallelization. The block factor that is used depends on how the memory is accessed for internal computations. The BRAM and AXI data port scheme employed in the developed DSM module is based on the example block diagram of Figure 3. If an argument of 4096 bytes has to be transferred to 4 BRAMs in the hardware kernel, then four AXI ports of 1024-bits width can be used in parallel to transfer 128 bytes of the argument in each BRAM at one clock cycle. The transfer of the whole argument will be completed in just 4096/(128 × 4) = 8 clock cycles. In case a single 1024-bit AXI bus port was used to transfer the argument in a single BRAM of 4096 bytes, then 4096/128 = 32 clock cycles would be required. The fastest implementation of the *Predict_kernel*() routine presented in this work employs two AXI bus ports of 1024 bits in order to transfer each one of the *split*1, *split*2 and *T_h_* thresholds arguments in a pair of independent BRAMs (i.e., six BRAMs are used in total to store these three arguments). An implementation with single ports per argument is also developed to compare the difference in the speed and the required resources. The ZynqMP Ultrascale+ FPGA does not provide enough resources to test more than two data ports of 1024-bit per argument.

As already described in the previous section, the sizes of split1 and split2 arguments (and the corresponding BRAMs) are 15,552 short integers of 16-bits and there are 15,552 threshold *T_h_* floating point values of 32-bits. If single wide ports of 1024-bits are used, then 1024/16 = 64 elements of split1 and split2 and 1024/32 = 32 elements of threshold values can be concurrently transferred in the three BRAMs of the hardware kernel. The time needed to transfer the split1, split2, and the thres arguments with single wide data ports, is 15,552/64 = 243 clock cycles and 15,552/32 = 486 clock cycles, respectively. If double ports are used for the split1, split2, and thres arguments, then the transfer of these arguments can be completed theoretically in half time. The BRAM of each argument will split into two components with equal size. In both cases (single and double data ports), the *N_c_* reference pixel intensities (e.g., 600 floating point numbers) are also transferred in a BRAM through a wide port of 800 bits that allow the concurrent transfer of 800/32 = 25 values in 600/25 = 24 clock cycles. Due to the small size of this argument, it is not split in multiple data ports. 

Xilinx provides coding techniques which force the synthesizer to implement an operation in hardware following specific rules. One of these coding techniques is the Resource pragma. This directive allows the use of specific resource types during implementation. More specifically, with this technique it was feasible to apply memory slicing using BRAMs for array partition instead of the Look Up Tables (LUTs) that the synthesizer uses by default. This is necessary since BRAMs with memory slicing show a lower access latency than LUTs. In this application, Dual Port BRAMs were employed to improve parallelization. 

To store the data in the BRAMs and take advantage of the parallelization, for-loops were used to store several elements of data in parallel in each iteration. To reduce latency during this procedure, the Pipeline pragma was used. This directive reduces the latency between consecutive Read or Write commands whenever possible. Pipeline is a technique that is used frequently at parallelization and in our approach, it was employed throughout Algorithm 1 (steps 2–8). The hardware acceleration HA2 that can be achieved using this technique cannot be theoretically estimated since it depends on the data dependencies between successive commands. However, if the average latency of each one of the successive commands is *L_p_* clock cycles, the acceleration HA2 is expected to be between 1 (no acceleration) and *L_p_* depending on the clock cycle where the successive commands get independent.

Another important acceleration technique was the selection of the appropriate communication protocol between the kernel and the host. It was important to calculate the size of the data that will be transferred from the host to the kernel and in the opposite direction in order to select the appropriate communication protocol. In this approach, an AXI master–slave interface was selected through the Interface directive. There is a set of optional and mandatory settings giving the flexibility to choose the mode of connection, the depth of the First-In-First-Out (FIFO) buffer that will be used during synthesis and implementation phase, the address offset etc. A different AXI-Lite interface was used for each argument of the hardware kernel. The FIFO depth varies depending on the size of the data that will be transferred through the AXI interface.

Finally, the last of the hardware acceleration techniques provided by OpenCL is the for-loop unrolling technique. If there are no dependences between the operations of the consecutive iterations, then unrolling a loop permits the execution of the operations of successive iterations in parallel. Of course, if a loop is fully unrolled, all of its identical operations are implemented with difference hardware circuits consuming a large number of resources. As an example, if we focus on the loop of 136 iterations that update the 68 pairs of face shape coordinates in step 8 of Algorithm 1, there is no dependence between these iterations. In the case of full for-loop unroll, all the 136 additions of double precision floating point numbers could be executed independently in one clock cycle, instead of a single addition per clock cycle. This technique is flexible since a partial loop unroll can be applied to make a tradeoff between resource allocation and speed. For example, in step 8 of Algorithm 1, an unroll factor *U_r_* = 8 is used, i.e., 8 double precision additions are performed in parallel in each clock cycle and 136/8 = 17 clock periods are needed to complete the whole step 8 instead of 136 clock periods that would be required if no loop unrolling were employed. Loop unrolling was also used for partial for-loop unrolling in step 7. The acceleration achieved with loop unrolling is equal to $HA3=U_{r}$.

Some for-loops in the original DEST code were difficult to unroll in an efficient way. The original DEST code was reconstructed in this case to achieve a better hardware acceleration. More specifically, the for-loop that traverses the regression trees in step 7 of Algorithm 1 was split manually in two independent but computationally equal loops (in steps 7a and 7b of Algorithm 1) that read their *split*1, *split*2, and *T_h_* threshold arguments from the pairs of the BRAMs where these arguments were stored in steps 4–6 of Algorithm 1. In this way, the Vitis hardware synthesis recognizes these two sections as independent and implements them in parallel. For instance, instead of running a for-loop that traverses 500 regression trees, two for-loops of 250 iterations are defined that are executed in parallel. This method can be extended to more AXI-bus ports and BRAMs as long as the FPGA resources can support the resulting number of ports.

### 3.4. Employed Rules for Increased Robustness

The first step in the analysis of a new input video frame is the detection of the face position using OpenCV library. As already mentioned in Section 3.2, this face detection does not necessarily have to be performed in all the successive input frames in order to save time and increase the overall frame processing throughput. For example, OpenCV library can be called to detect the position of the face once every five frames. If shape alignment has to be performed in all the intermediate frames, the last bounding box returned by OpenCV can be slightly extended, assuming that the face has not moved outside this extended bounding box in such a short time interval. As can be deduced, the successful shape alignment is heavily dependent on the face position recognition of the OpenCV. Low-resolution, noisy images with dark lighting conditions can often lead to OpenCV failure in the correct detection of the position and the dimensions of the face bounding box. The estimation of the distances shown in Figure 1 and the recognition of yawning and eye blinks that are based on these distances can be disrupted by OpenCV failures or by an inaccurate shape estimation performed by the ERT model. 

Before applying the following frame-to-frame coherency rules that have been incorporated in the top level code of the developed DSM module, a valid bounding box and face shape have to be recognized as reference. This is performed by confirming that the bounding box and face shape size and position have coherence in the last *F_r_* frames. The reference bounding box is updated with a new bounding box provided that it does not violate any of the rules that are defined in this section. Let (*Rows*, *Cols*) be the resolution of the image frame. Let also (*BX_r_*_1_, *BY_r_*_1_) and (*BX_r_*_2_, *BY_r_*_2_) be the coordinates of the top left and bottom right corner of the reference bounding box; (*CX_r_*_1_, *CY_r_*_1_) and (*CX_r_*_2_, *CY_r_*_2_) are the coordinates of the top left and bottom right corner of the current bounding box, while (*SX_r_*_1_, *SY_r_*_1_) and (*SX_r_*_2_, *SY_r_*_2_) are the upper eyebrow and the lower chin landmark used to estimate the shape height *H_f_*:(12)$$H_{f}=\sqrt{{\left({SX}_{r1}-{SX}_{r2}\right)}^{2}+{\left({SY}_{r1}-{SY}_{r2}\right)}^{2}}$$

#### 3.4.1. Head Bounding Box Absolute Dimension Restrictions Related to the Frame Size

The first rule concerns the size of the bounding box returned by OpenCV. Since the camera is placed inside a car and its distance from the driver is relatively short and constant, the absolute size of the bounding box cannot be smaller than a minimum fraction *PX_min_* from the frame width (*Cols*) and a minimum fraction *PY_min_* from the frame height (*Rows*). In our experiments, ${PX}_{min}={PY}_{min}=20\%$. In a more formal way, the following conditions should be true:(13)$$\left|{CX}_{r1}-{CX}_{r2}|>{|PX}_{min}\cdot Cols\right|$$
(14)$$\left|{CY}_{r1}-{CY}_{r2}|>|{PY}_{min}\cdot Rows\right|$$

If one of the relations (10) or (11) is not true, then the returned bounding box is assumed invalid and the reference bounding box is used in the current frame potentially extending its size by e.g., 5%. However, if too many successive frames are rejected, the reference bounding box is defined from scratch because the available one is too old to be considered valid. In our experiments the reference bounding box is reset if 10 consecutive frames are rejected.

An unacceptable bounding box that leads to a false shape prediction is shown in Figure 4.

#### 3.4.2. Coherency in the Head Bounding Box Dimensions

The second rule compares the size of the current bounding box to the size of the reference one. It should neither be larger, nor smaller than a fraction *PX_ref_* of the reference width and a fraction *PY_ref_* of the reference height. In our experiments: ${PX}_{ref}={PY}_{ref}=30\%$.
(15)$$\left(1+{PX}_{min})\cdot |{BX}_{r1}-{BX}_{r2}|>|{CX}_{r1}-{CX}_{r2}|>(1-{PX}_{min})\cdot |{BX}_{r1}-{BX}_{r2}\right|$$
(16)$$\left(1+{PY}_{min})\cdot |{BY}_{r1}-{BY}_{r2}|>|{CY}_{r1}-{CY}_{r2}|>(1-{PY}_{min})\cdot |{BY}_{r1}-{BY}_{r2}\right|$$

If at least one of the relations (12) or (13) is not valid, the current OpenCV bounding box is rejected and a similar procedure is followed as in the previous rule. Figure 5 shows an example where this rule is violated.

#### 3.4.3. Coherency in the Head Bounding Box Position

Although the size of the detected bounding box may not violate the previous two rules, its position may deviate excessively from the position of the reference bounding box as shown in Figure 6. It is expected that the driver’s face cannot have moved too far away from its reference position between successive frames. The maximum allowed deviation can be expressed as the *P_dev_* fraction (e.g., *P_dev_* = 30%) of the reference bounding box width (or height, or both): (17)$$\left|{BX}_{r1}-{CX}_{r1}\right|<P_{dev}\cdot |{BX}_{r1}-{BX}_{r2}|$$

If relation (14) is violated, the OpenCV bounding box is rejected and a similar procedure is followed as in the previous rules. 

#### 3.4.4. Face Shape Size Restrictions

All previous rules refer to the bounding box position and size. However, even if the current bounding box is acceptable, the face shape alignment within this box may fail as shown in Figure 7. This last rule compares the face shape height *H_f_* to the height of the reference bounding box and this height should not be less than a fraction *P_smin_* (e.g., 30% similarly to *PX_ref_*, *PY_ref_*) of the reference bounding box height.
(18)$$H_{f}>P_{smin}\cdot |{BY}_{r1}-{BY}_{r2}|$$

## 4. Results

The experimental results presented in this section have been retrieved using the Xilinx ZCU102 FPGA development board with the PL operating at a clock frequency of 150 MHz. Four ERT models have been trained differing in the parameters shown in Table 1. As already described in Section 3.2, the default ERT model (called M0) consists of *T_c_* = 10 cascade stages, the sparse representation of the input image consists of *N_c_* = 600 reference pixels, each cascade stage has *N_tr_* = 500 regression binary trees, and each tree has a depth of *T_d_* = 5, i.e., 31 nodes. The ERT model M1 differs from the default in the number of reference pixels since it defines *N_c_* = 800. With a larger number of reference pixels, we expect to achieve a higher precision, since the image representation is more accurate. The ERT models M2 and M3 differ from the default in the number of regression trees in each cascade stage. M2 has fewer regression trees; thus, we expect a faster operation but with a potential penalty in the accuracy, while M3 has more regressor trees than the default. Testing M3 will reveal if the increase in the number of regression trees will lead to a higher precision with a penalty in the execution time. 

The application dest_train of the DEST package was used to train the ERT models M0–M3. This application displays information about the average error measured in each cascade stage of the training algorithm. This error is the average of the normalized distance between the landmarks estimated during training and the annotated ground truth. The evolution of the average error per cascade stage for each one of the models M0 to M3 is shown in the diagram of Figure 8. As can be seen from this figure, the error is reduced in less than half in the first four cascade stages, while in the last cascade stages an error floor appears between 1.5% and 1.8%, depending on the model. For this reason, there is no benefit in defining more cascade stages since the landmark prediction time is increased proportionally without significant reduction in the average error.

Each ERT model is supported by a different hardware kernel that will be called with the same name for simplicity (kernel M0, M1, M2, M3). The kernel M0 is implemented both with one wide port per argument as well as with two wide ports per argument in order to compare the acceleration achieved with two wide ports as well as the difference in the required resources and power consumption. Accuracy is not affected by the number of wide ports used per argument. 

The different parameters of the models M0-M3 (listed in Table 2) affect the FPGA resources and the execution time of the landmark alignment algorithm as shown in Table 3. The resources and the latency for processing a single frame is listed in Table 3 for the different hardware kernels that support each ERT model. The resources listed in Table 3 are Look-Up Tables (LUT), LUT Random Access Memory (LUTRAM), Flip Flops (FF), Block RAM (BRAM), and Digital Signal Processing (DSP) units. The latency for processing a single frame is approximately *T_c_* times (number of cascade stages, e.g., *T_c_* = 10 in all models) the latency of a single hardware kernel execution. Only a small fraction of this latency is spent in the preparation of the kernel input/output parameters since all of the large buffers that are kernel arguments (split1, split2, thres) in each cascade iteration have already been prepared during initialization (in the *predict_prepare*() routine)**.**

The speed achieved with each {ERT model, supporting hardware kernel} pair, measured in fps, is compared to some referenced approaches in Table 4. In this table, the shortest frame processing latency achieved by the default model M0 on an Ubuntu i7 platform without the latency of the employed rules for higher robustness is also listed for comparison with the Local Binary Features (LBF) approach presented in [1]. In Table 4, the resolution of the images and the number of landmarks aligned is also listed for a fair comparison.

To measure the precision of each model in the recognition of yawning, 30 videos were used from YawDD [36] and 30 from NITYMED [37] datasets. In each dataset, the number of male drivers is equal to the number of female drivers. In each video from NITYMED suite, the driver yawns three times while in each video in YawDD, the driver yawns 2–4 times. The True Positives (TP), False Positives (FP) and False Negatives (FN) can be measured from the output videos that are stored in the ZCU102 memory card after processing an input video. True Negatives (TN) cannot be measured because the number of TN samples cannot be estimated from the video in periods when no TP, FP or FN is recognized. It is easy to measure TN samples in single images when there is no yawning but not in a video where the yawning is recognized when the driver’s mouth is found open for a minimum period of time. Therefore, the metrics employed are the sensitivity and precision that can be estimated using TP, FP, and FN.
(19)$$Sensitivity=\frac{TP}{\begin{array}{c} \mathrm{TP}+\mathrm{FN} \end{array}}$$
(20)$$Precision=\frac{TP}{TP+FN}$$
(21)$$Accuracy=\frac{TP+TN}{TP+TN+FP+FN}$$

The specificity cannot be estimated either, since it also depends on the TN samples. The sensitivity and precision achieved for each ERT model are presented in Table 5. The accuracy of the referenced approaches that detect yawning is also listed in Table 5.

Concerning the dynamic power consumption of the Programmable Logic (PL) part of the FPGA, the model M0 draws 1.18 W when implemented with a single wide port per argument, while this power is in the order of 2.7 W for all the models implemented with a double wide port per argument.

## 5. Discussion

In Table 3 presented in Section 4, the default model was implemented both with single and double wide ports per argument (concerns the arguments split1, split2, and thres, as discussed in Section 3.3). The FPGA resources needed in the implementation with single wide port per argument are almost half of those needed by the implementation with double wide ports. This is due to the fact that most of the resources are required for the implementation of the AXI interface ports and the BRAMs where the arguments are transferred in order to allow the hardware kernel to access them locally. The number of resources needed for the implementation of the Regressor::predict() and Tree::predict() functionality and the passing of the rest of the arguments is by far smaller than those required to pass the three arguments through double wide ports. It is roughly estimated that about 44 K LUTs are used for the wide port of the split1, split2, and thres arguments in the single wide port implementation, the corresponding BRAMs where these arguments will be stored in the kernel and the implementation of a single loop in the step 7 of Algorithm 1. About 10 K LUTs are allocated for passing the rest of the arguments and the remaining functionality of the *predict*() functions. The LUT resources for the double wide ports implementation is about 88 K for passing the arguments split1, split2, and thres and for the implementation of the pair of loops shown in step 7 of Algorithm 1. The LUTs required for passing the rest of the arguments remain in the order of 10 K. The benefit from allocating a much higher number of FPGA resources (more than 140%) results in a significant hardware acceleration since the kernel latency is reduced from 46 ms to less than 20 ms when double wide ports are employed per argument. 

The rest of the kernels (M1, M2, M3) described in Table 3 were implemented using double wide ports per argument since the main target of this work is to achieve the maximum processing speed. The resources required by the kernels M1, M2, M3 are almost the same, confirming that the major part of the allocated FPGA resources is used for the implementation of the double wide ports. Some minor reduction in the LUTRAMs and FFs in kernels M2 and M3 are not related to a specific parameter of Table 2. It is caused by the modifications needed to handle a variable number of regressor trees. Model M1 differs from the default M0 only in the number of reference pixels intensities and thus, its resources are almost identical to those required by M0. Their latencies are also comparable. Although a larger number of pixel intensities has to be transferred to the kernel, this data copy happens in parallel with the transfer of the larger arguments (split1, split2, and thres). M2 and M3 differ from the default M0 in the number of regression trees visited in each cascade stage. Therefore, their latency listed in Table 3 is proportional to the number of trees. Their resources do not differ significantly.

From the comparison presented in Table 4, it is obvious that the achieved frame processing speed is much higher than the related approaches. More specifically, the fastest approach [1] has a frame processing latency ranging between 1.1 ms and 7.2 ms. However, if this is compared to our implementation on an Ubuntu i7 platform (without the application of the rules that increase robustness), our software acceleration method achieves a latency less than 40% of the lowest latency achieved in [1]. The face alignment applications ([8,9,10,12]) based on ERTs [14] achieve a relatively high speed (between 16 and 45 fps) but they concern different applications such as face recognition, pose estimation, etc., and some of them (e.g., [9]) align a smaller number of landmarks, which is a faster procedure. The yawning detection approaches [30,32] are based on CNNs and operate at a significantly smaller speed. The DAN approach of [33] that is also based on CNNs achieves a comparable speed to our implementation on the embedded platform ZCU102 but this speed is measured in [33] on a GPU platform (GeForce GTX1070). The frame processing speed of our approach does not depend on the image resolution since it is applied on a sparse representation of the input image consisting of a constant number of *N_c_* reference pixels.

In Table 5, the success rate in the recognition of yawning is compared between the proposed and the related approaches. Although the measured sensitivity and precision are not directly comparable with the accuracy reported in the related approaches, it is obvious from Table 5 that the yawnings are recognized with a better success rate in our method compared to almost all the referenced approaches (only [31] can achieve comparable success rate with M2). The error measured in [1] and [33] can be compared to the training error presented in Figure 8 which is less than 2% in the 10th cascade stage of each model. Of course, this is an error extracted from the training set of photographs. If a similar error is extracted from a different test set, the average error is in the order of 5% which is comparable or smaller than the errors presented in [1] and [33]. The highest sensitivity is achieved by M1 (92.3%) and the highest precision by M2 (96.6%). This stems from the fact that with M1 we get a smaller number of false negatives while with M2 we get a smaller number of false positives. The use of a higher number of reference pixels as in M1 seems to favor sensitivity. Comparing the use of a larger number of regression trees as in M3, it improves slightly the sensitivity (85% in M3, 82.15% in M2) but with a much worse precision compared to M2 that is based on a smaller number of regression trees (precision is 96.6% in M2 and 70.8% in M3). These results show that increasing the number of regressor trees above 400 does not further improve the precision achieved in the recognition of yawning, although the training error in Figure 8 is reduced when a larger number of regressor trees are used.

Figure 8 can be used as a rule of thumb concerning the tradeoff between speed and accepted training error since the frame processing latency is proportional to the number of cascade stages and the number of regressor trees/cascade stage. The frame processing latency *L_f_* (without the rules for increase robustness) of the default model M0 (*N_tr_* = 500, *T_cs_* = 10) can be adapted to $L_{f}^{\prime }$ according to the following equation if a model with different number of trees $N_{tr}^{\prime }$ and cascade stages $T_{cs}^{\prime }$ is selected:(22)$$L_{f}^{\prime }=\frac{T_{cs}^{\prime }N_{tr}^{\prime }}{T_{cs}N_{tr}}L_{f}$$

For example, if a training error of 2% is acceptable, one of the models M0, M1, or M3 can be employed with $T_{cs}^{\prime }=7$ cascade stages. If M3 ($N_{tr}^{\prime }=600$) is used for lower error, the frame processing speed becomes $L_{f}^{\prime }=\frac{7\cdot 600}{10\cdot 500}L_{f}=0.84\cdot L_{f}$.

Concerning the power consumption of the various FPGA components, a significantly smaller dynamic power consumption of the PL is achieved with the single wide port implementation of M0 kernel. More specifically, the single wide port implementation of M0 consumes 1.181 W compared to the 2.781 W consumed by the double wide port implementation of this kernel. The dynamic power of the rest of the kernels is almost identical to that of the M0 with double wide ports. The effect of the frame-to-frame coherency rules presented in Section 3.4 is significant but cannot be easily quantified since a large number of face recognition failures occur, especially in the nighttime videos of the NITYMED dataset. When PERCLOS is used for yawning detection, the shape of the open mouth has to be recognized in several successive frames. Without the coherency rules, open mouth shape is not recognized in many intermediate frames, leading to a failure in recognizing the yawning. It can be roughly estimated that half of the yawning in the nighttime videos cannot be recognized if the coherency rules are deactivated.

Summarizing the conclusions from the experimental results, we can state that the advantages of the proposed DSM module are the following: (a) high frame processing speed reaching 63 fps due to the acceleration that has been achieved both at the level of software restructuring (elimination of time-consuming Eigen calls) and at hardware acceleration level with kernels based on double wide ports per argument, (b) employing frame-to-frame coherency rules allows a robust operation and a higher precision in yawning detection that can reach up to 96.6%, (c) the proposed face alignment scheme can also be used to detect sleepy eye blinks, microsleep, and driver distraction, (d) the ERT model parameters that can improve the precision and speed have been identified (e.g., higher number of reference pixels improve sensitivity but larger number of regressor trees does not improve precision), (e) the DSM system is available for Ubuntu environment (mainly for debug reasons and interactive improvement of the algorithms and techniques that are developed) as well as for embedded target platforms such as Xilinx ZCU102, (f) the new video dataset with night-time, real car environmental conditions that has been created to evaluate the developed DSM module is available to the public. The hardware acceleration techniques (e.g., the use of double wide ports per argument) discussed in Section 3.3 can be useful to other computationally intensive applications that require the transfer of a large amount of data to the hardware kernel.

## 6. Conclusions

In this paper, a Driver State Monitoring (DSM) module has been developed based on face shape alignment using a machine learning technique called Ensemble of Regression Trees. The Deformable Shape Tracking (DEST) library has been ported both to the Ubuntu i7 host computer and an embedded Xilinx ZCU102 FPGA board. A significant acceleration has been achieved both at the software and the hardware level, resulting in a frame processing speed higher than 60 frames/s with a remarkable precision in yawning recognition that exceeds 96%. Frame-to-frame coherency rules have been defined to achieve a robust operation in night-time conditions when other image processing methods fail to operate accurately. A new public dataset has been created to evaluate the developed DSM module in night-time scenarios and in a real moving-car environment. The employed hardware acceleration techniques, such as the use of double wide data ports per argument, can be exploited by different applications where the computationally intensive operations also need the transfer of large amount of data to the hardware kernel.

Future work will focus on the further extension and optimization of the rules for increased robustness and the dynamic selection of an appropriate shape alignment rate. Further improvements in the hardware acceleration techniques will also be investigated. Finally, the use of the developed DSM module for other applications (microsleep detection, driver distraction, face expression recognition, etc.) will also be examined.

## Figures and Tables

**Figure 1 sensors-23-06344-f001:**
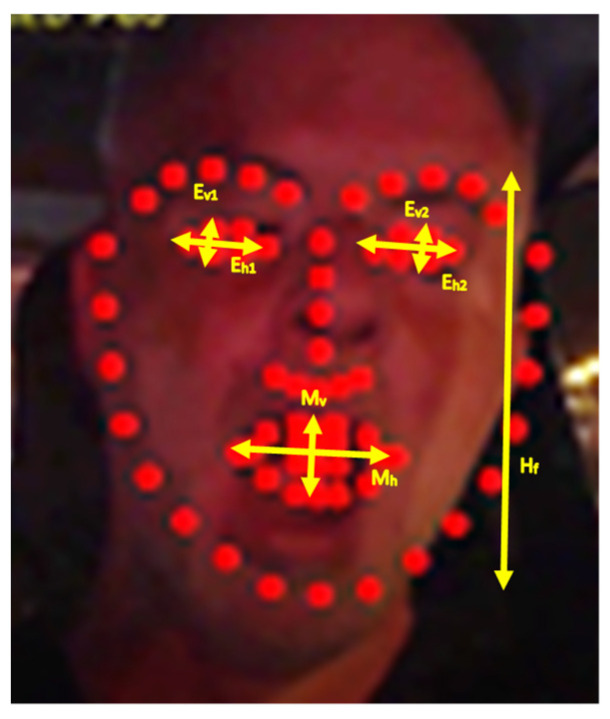
Alignment of 68 landmarks on a driver’s face in nighttime conditions.

**Figure 2 sensors-23-06344-f002:**
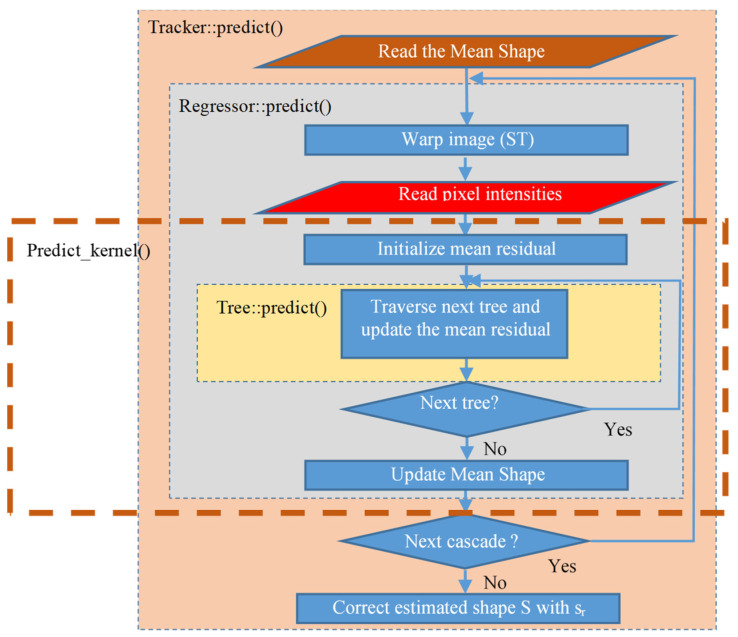
The operations in the landmark predict() functions of the DEST video tracking application.

**Figure 3 sensors-23-06344-f003:**
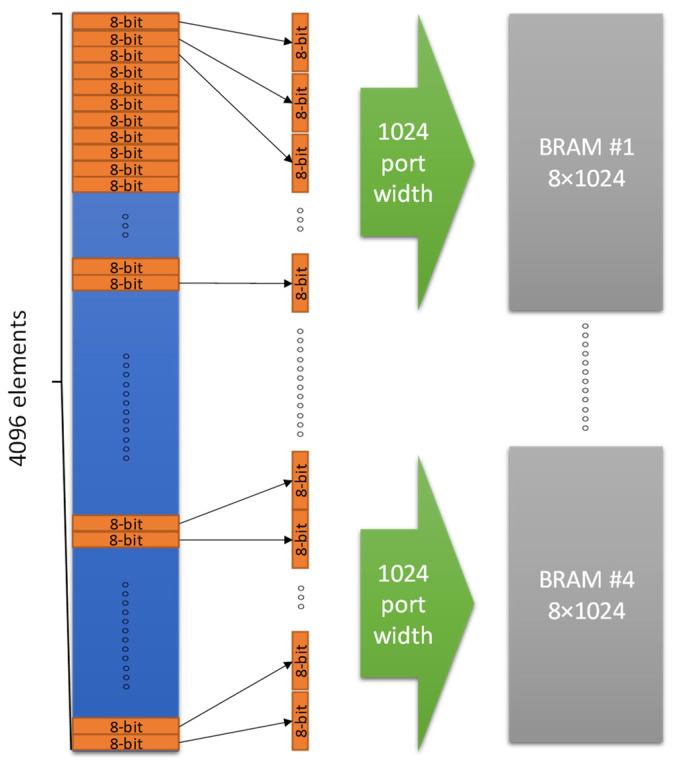
AXI wide ports and BRAMs used to transfer hardware kernel arguments.

**Figure 4 sensors-23-06344-f004:**
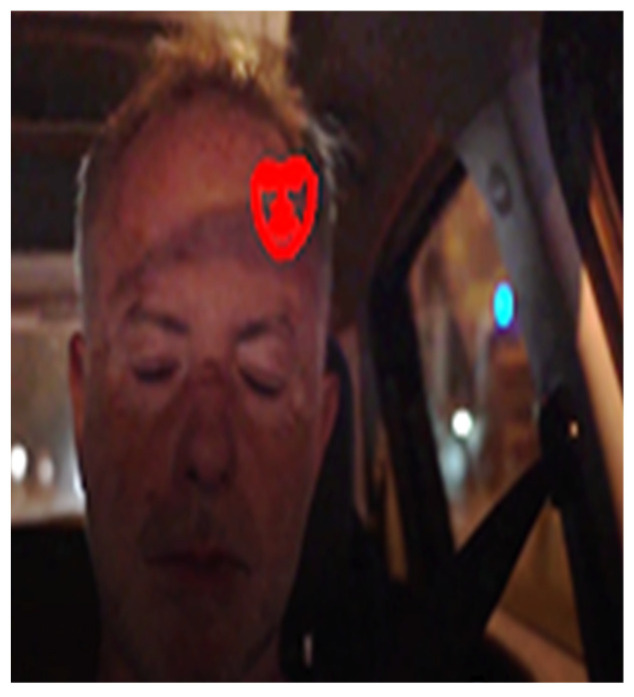
Violation of the absolute dimensions of the shape according to relations (10) and (11).

**Figure 5 sensors-23-06344-f005:**
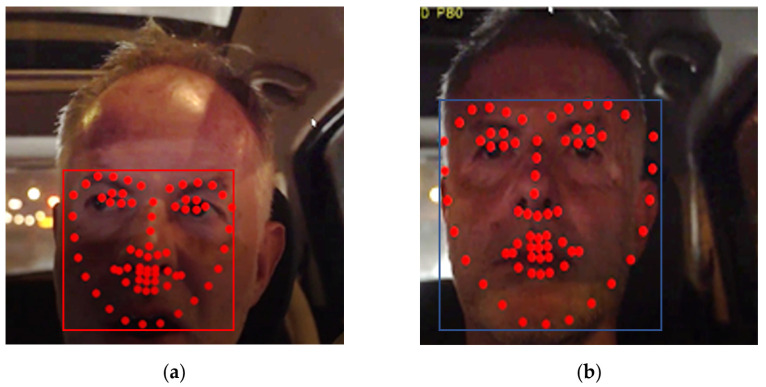
The current bounding box (**a**) violates relations (12) and (13) when compared to the reference bounding box (**b**).

**Figure 6 sensors-23-06344-f006:**
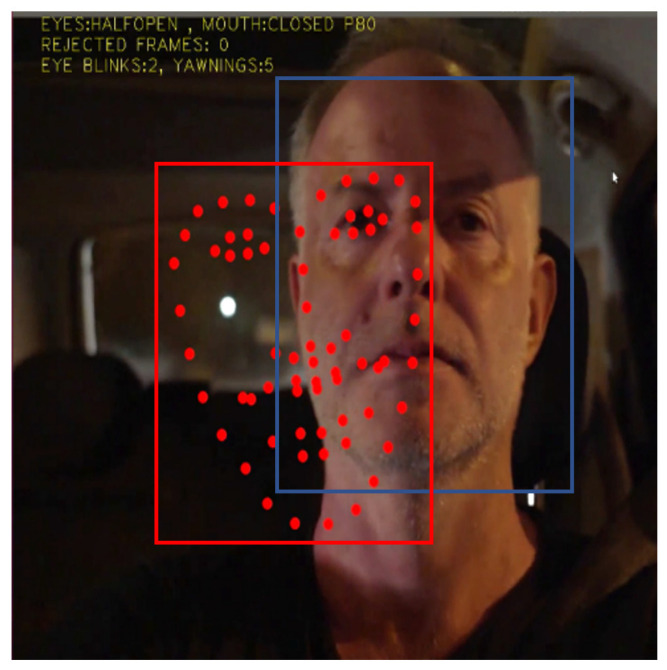
The current bounding box (red) position should not exceed a maximum distance from the reference bounding box (magenta).

**Figure 7 sensors-23-06344-f007:**
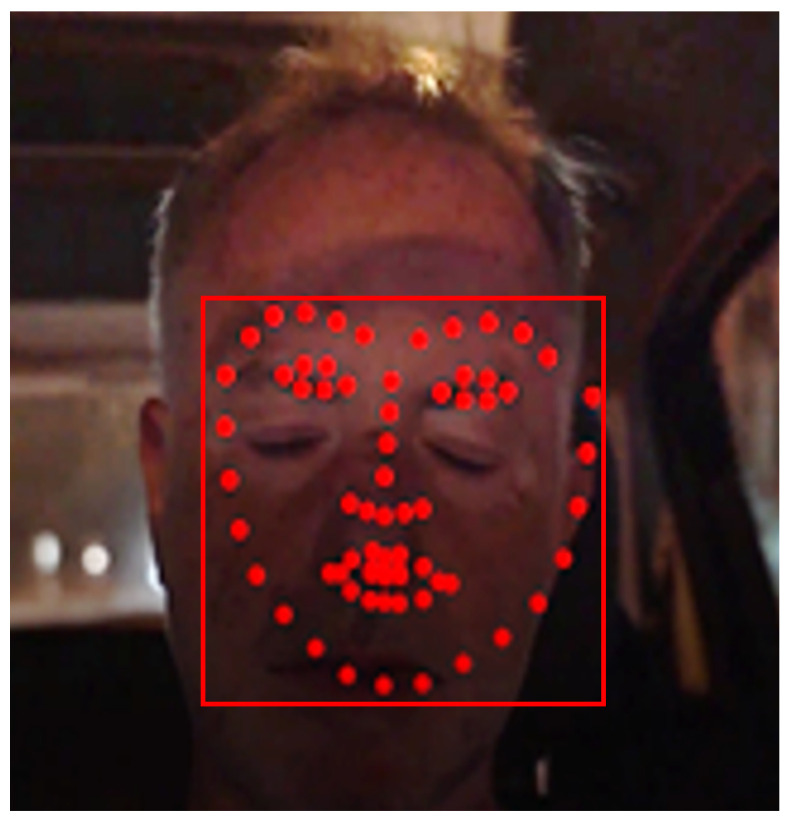
Valid bounding box with invalid shape alignment.

**Figure 8 sensors-23-06344-f008:**
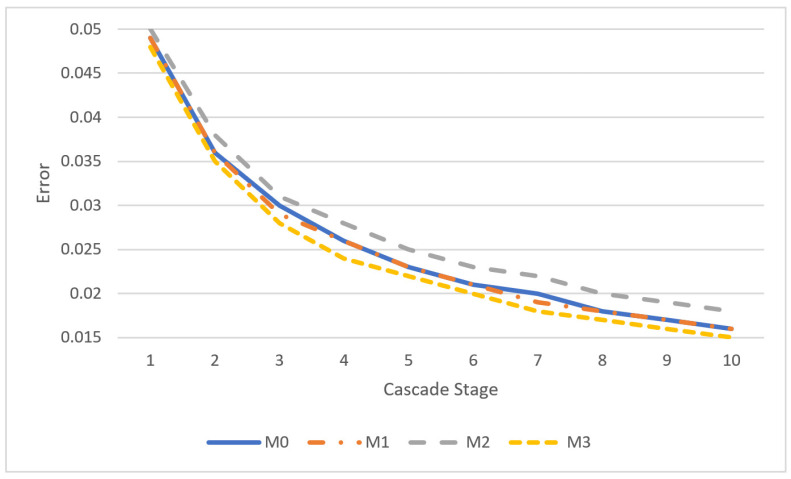
Average normalized error per cascade stage for each model.

**Table 1 sensors-23-06344-t001:** The employed hardware acceleration techniques.

Acceleration Technique	Justification
Wide Ports	High-speed transfer of large arguments to the HW kernel.
Double, Wide Ports Per Argument	To transfer concurrently parts of big arguments to the HW kernel.
Local BRAM	To store locally the arguments transferred through wide ports and access them with high speed from the HW kernel.
Dataflow	To exploit parallelism in successive C commands if they do not have data dependencies.
Loop Unrolling	To reduce loop iterations.

**Table 2 sensors-23-06344-t002:** ERT Models.

Model	Wide Ports per Argument	Trees (*N_tr_*)	Reference Pixels (*N_c_*)
Model M0 (single wide port)	1	500	600
Model M0 (double wide port)	2	500	600
Model M1 (double wide port)	2	500	800
Model M2 (double wide port)	2	400	600
Model M3 (double wide port)	2	600	600

**Table 3 sensors-23-06344-t003:** FPGA resources and latency for single frame processing. Kernels M1, M2, M3 have been implemented with double wide ports per argument while the default kernel M0 has been implemented both with single and double wide ports per argument.

Resources	M0 (Single Wide Port per Argument)	Model M0 (Single Wide Port)	Model M1 (Double Wide Port)	Model M2 (Double Wide Port)	Model M3 (Double Wide Port)
LUT	54,149 (19.76%)	98,776 (36.06%)	98,776 (36.04%)	98,145 (35.81%)	98,145 (35.81%)
LUTRAM	2862 (1.99%)	6205 (4.31%)	6205 (4.31%)	5765 (4%)	5765 (4%)
FF	84,221 (15.36%)	168,067 (30.66%)	168,067 (30.66%)	166,400 (30.36%)	166,433 (30.36%)
BRAM	139 (15.19%)	652 (71.55%)	653 (71.55%)	653 (71.55%)	653 (71.55%)
DSP	40 (1.59%)	56 (2.22%)	56 (2.22%)	56 (2.22%)	56 (2.22%)
Processing Latency	46.632 ms	19.881 ms	21.367 ms	15.881 ms	22.956 ms

**Table 4 sensors-23-06344-t004:** Comparison of frame processing speed with referenced approaches.

Reference Application	Frame Processing	Number of Landmarks	Image Resolution
[1] (Face Recognition based on LBF)	1.1 ms to 7.2 ms	68	1920 × 1080
[8] (Face Recognition)	45 fps	-	20 × 20
[9] (Pose Estimation)	30 fps	23	320 × 240
[10] (Head Pose Estimation)	30 fps	68	-
[12] (3D Face Alignment)	16 fps	68	640 × 480
[30] (Yawning classification)	3 fps	CNN (no landmarks)	640 × 480
[32] (Driving Drowsiness)	4.33 fps	CNN (no landmarks)	100 × 100
[33] (Face Recognition on a GeForce GTX 1070 GPU)	45 fps	68	1920 × 1080
M0 Model on Intel i7 (no robustness rules)	0.45 ms	68	640 × 480 to 1920 × 1080
M0 Model on ZCU102	50 fps	68	640 × 480 to 1920 × 1080
M1 Model on ZCU102	47 fps	68	640 × 480 to 1920 × 1080
M2 Model on ZCU102	63 fps	68	640 × 480 to 1920 × 1080
M3 Model on ZCU102	43 fps	68	640 × 480 to 1920 × 1080

**Table 5 sensors-23-06344-t005:** Sensitivity, precision, and accuracy of the developed ERT models and the referenced approaches that detect yawning.

Reference	Success Rate	Metric
[1] (LBF)	8.2–25.5%	Landmark position error
[30]	65–75%	Accuracy
[31]	80–98%	Accuracy
[32]	64%	Accuracy
[33]	3.89–5.03%	Landmark position error
M0 Model	75.75%	Sensitivity
	80.6%	Precision
M1 Model	92.3%	Sensitivity
	85.7%	Precision
M2 Model	82.85%	Sensitivity
	96.66%	Precision
M3 Model	85%	Sensitivity

## Data Availability

Dataset available at IEEE Dataport [37].

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
