# Peer review of "A High Performance and Robust FPGA Implementation of a Driver State Monitoring Application"

_sensors, 2023, doi:10.3390/s23146344_

Round 1

Reviewer 1 Report

1、 It is suggested that the authors add a review of recent face shape alignment methods and summarize machine learning methods other than ERT.

2、 It is suggested that the authors explain why they chose ERT as the model instead of other methods.

3、 The presentation of experimental results is too simple, and the authors should consider enriching the presentation of experimental results.

4、 The authors should discuss and explain in depth the experimental results obtained, and analyze other factors that may affect system performance and accuracy, such as lighting, occlusion, and posture.

5、 The article lacks theoretical proof and analysis of the proposed hardware acceleration technology, and it is suggested that the authors add relevant discussions on specific hardware design and implementation details.

Author Response

Dear reviewer,

We would like to thank you in advance for the time you spent reading out our manuscript. Your comments were valuable and based on them we have addressed all the points you mentioned. We strongly believe that the paper meets the high-quality criteria of MDPI Sensors journal. The modifications are highlighted in yellow in the revised manuscript. Below you can find our responses to each comment along with references to the section, paragraph, sentence, or lines where you can find the additional or modified text that addresses your concerns. We are at your disposal for any other modification required.

Best Regards

Nikos Petrellis

On behalf of all the authors.

Comment 1

It is suggested that the authors add a review of recent face shape alignment methods and summarize machine learning methods other than ERT.

Author Response 1

Thank you for your comment. Two seminal approaches were added (references [1] and [33]) for comparison. Moreover, references [5-6] and [16] are also added. Please see these references and the sentences 1, 5, 6, 7 of paragraph 1 in Introduction section, last sentence of paragraph 5 in Introduction section, extended Tables 4 and 5, and extended paragraphs 3-6 in Discussion section.

Comment 2

It is suggested that the authors explain why they chose ERT as the model instead of other methods.

Author Response 2

Thank you for your comment. ERT was selected due to the high speed achieved with this method as well as the fact that it has been employed in various popular image processing libraries (DLIB and Deformable Shape Tracking (DEST)). Moreover, open-source code for the implementation of this method was also available (DEST repository). Please see the modified last 3 sentences (lines 59-63) of the 2nd paragraph of Introduction section. The referenced DSM systems are implemented with low-speed face alignment methods, are tested mainly in daytime and depend on face recognition services that can lead to loss of focus and failures. Our approach aims to overcome these issues and offer acceleration and robust operations solutions useful for other approaches based on ERTs. Please see the new 4 sentences at the start of paragraph 6 in Introduction section (lines 110-118).

Comment 3

The presentation of experimental results is too simple, and the authors should consider enriching the presentation of experimental results.

Author Response 3

Thank you for your comment. In Section 4 the experimental results are introduced in the form of Tables and Figures but they are actually discussed in Section 5. Both tables, figures and their descriptions have been extended. Please check the new figure 8 that depicts the average normalized error/ cascade stage for every model, the extended tables 4 and 5 which are enriched with new references and the extended paragraphs 3-6 in the Discussion section.

Comment 4

The authors should discuss and explain in depth the experimental results obtained, and analyze other factors that may affect system performance and accuracy, such as lighting, occlusion, and posture.

Author Response 4

Thank you for your comment. The original ERT method presented in [14] is a 2D shape alignment method that can be successfully applied to shapes with tilt less than +/-20 degrees. It also does not support faces with occlusion (sunglasses, faces obscured by other objects etc.). The aim of the proposed approach is to offer speed and robustness enhancements rather than extend the ERT method to solve these issues. However, the proposed acceleration and shape alignment validation techniques can be applied without any modifications to extensions of [14] that align 3D shape [17] or handle occlusions [20]. Please check the new last 3 sentences of the 2nd paragraph of Introduction section (lines 59-63), the extended-modified 3rd paragraph of Introduction section (lines 65-85), and the extended paragraphs 3-6 in Section 5.

Comment 5

The article lacks theoretical proof and analysis of the proposed hardware acceleration technology, and it is suggested that the authors add relevant discussions on specific hardware design and implementation details

Author Response 5

Thank you very much for your comment. Section 3.3 is modified and the theoretical estimation of the following hardware acceleration techniques are estimated in a general manner: wide ports (HA1), pipeline (HA2) and loop unrolling (HA3). In the Discussion section a formal method has also been added to estimate the frame processing latency of custom ERT models. Please see the new equation (9)-(11) and the corresponding highlighted text (lines 417-425), the new last 2 sentences of paragraph 9 (lines 485-490) in section 3.3, the new last sentence of paragraph 11 (lines 515-516) in section 3.3, and the new paragraphs 5-6 (lines 755-763) of Section 5 and the new equation (22).

Reviewer 2 Report

This research paper presents a high-performance Driver State Monitoring (DSM) application that uses an Ensemble of Regression Trees (ERTs) machine learning method to detect driver drowsiness. The application focuses on accelerating frame processing using reconfigurable hardware, reducing latency and improving robustness. The DSM application achieves a frame processing rate of up to 65 frames per second with high sensitivity (93%) and precision (97%) in yawning recognition. Unconventional hardware acceleration techniques are employed to address the challenges of implementing the DSM algorithm in reconfigurable hardware, which may have broader applications beyond this specific domain. Followings are my concerns:

1. The key findings and contributions of this paper should be succinctly summarized and effectively showcased.

2. The rationale behind introducing the new approach should be presented with enhanced clarity and emphasis.

3. To show the importance of the research topic, more related works can be reviewed. For example, High Performance FPGA Implementation of Single MAC Adaptive Filter for Independent Component Analysis.

4. In order to enhance the quality of the figures, it is recommended to make improvements to their resolution, ensuring clearer and more detailed visual representation.

5. In comparison to other approaches, what are the distinguishing advantages of the proposed approach?

Minor editing of English language required.

Author Response

Dear reviewer

We would like to thank you in advance for the time you spent reading out our manuscript. Your comments were valuable and based on them we have addressed all the points you mentioned. We strongly believe that the paper meets the high-quality criteria of MDPI Sensors journal. The modifications are highlighted in yellow in the revised manuscript. Below you can find our responses to each comment along with references to the section, paragraph, sentence, or lines where you can find the additional or modified text that addresses your concerns. We are at your disposal for any other modification required.

Best Regards

Nikos Petrellis

On behalf of all the authors.

Comment 1

The key findings and contribution of this paper should be succinctly summarized and effectively showcased.

Author Response 1

We appreciate your comment. The key findings and contributions of this paper are as follows. First, the Hardware and Software acceleration techniques which have been applied to the ERT face alignment open-source code DEST. Secondly, the significant reduction of the frame processing latency allows a higher accuracy without sacrificing speed that can reach 65 fps. In addition, face alignment result validation rules that increase the robustness and the accuracy of the system have been employed. We continue with unconventional hardware acceleration techniques that have been used and can also be employed in the acceleration of other applications that require the transfer of large volume of data to their hardware kernels. Finally, a new dataset has been developed to train and test facial shape alignment models in nighttime conditions. These contributions were clearly defined in the last paragraph of Section 5 (lines 777-794) but the Abstract and Introduction were also modified to present these contributions more clearly. Please see the modified 2nd, 3rd and last sentence in the Abstract and also the new 3rd paragraph before the end of Introduction section (lines 152-161)

Comment 2

The rationale behind introducing the new approach should be presented with enhanced clarity and emphasis.

Author Response 2

Thank you for your comment. ERT was selected due to the high speed achieved with this method as well as the fact that it has been employed in various popular image processing libraries (DLIB and Deformable Shape Tracking (DEST)). Moreover, open-source code for the implementation of this method was also available (DEST repository). Please see the modified last 3 sentences (lines 59-63) of the 2nd paragraph of Introduction section. The referenced DSM systems are implemented with low-speed face alignment methods, are tested mainly in daytime and depend on face recognition services that can lead to loss of focus and failures. Our approach aims to overcome these issues and offer acceleration and robust operations solutions useful for other approaches based on ERTs. Please see the new 4 sentences at the start of paragraph 6 in Introduction section (lines 110-118).

Comment 3

To show the importance of the research topic, more related works can be reviewed. For example, High Performance FPGA Implementation of Single MAC Adaptive Filter for Independent Component Analysis

Author Response 3

Thank you for your comment. Two seminal approaches were added (references [1] and [33]) for comparison. Moreover, references [5-6] and [16] are also added. Please see these references and the sentences 1, 5, 6, 7 of paragraph 1 in Introduction section, last sentence of paragraph 5 in Introduction section, extended Tables 4 and 5, and extended paragraphs 3-6 in Discussion section.

Comment 4

In order to enhance the quality of the figures, it is recommended to make improvements to their resolution, ensuring clearer and more detailed visual representation.

Author Response 4

Thank you very much for your suggestion. These figures were on purpose containing low resolution images in nighttime to illustrate the case in which the DSM performs under worst case operating conditions. To be clearer we increased the brightness of the figures 1, 4, 5, 6, and 7.

Comment 5

In comparison to other approaches, what are the distinguishing advantages of the proposed approach?

Author Response 5

Thank you very much for your comment. Compared to Deep Learning approaches, much higher speed is achieved as can be seen from Table 4. Also, robustness due to the frame coherency rules applied increases precision as you can see in Table 5. In addition, the latency caused of C++ Eigen library is reduced and there is a speed-up with software restructure as you can see in paragraph 7, sentences 2-3 of the Introduction section (lines 137-144). Finally, unconventional hardware acceleration techniques, exploitable from other examples, where used including large buffers that used and have to be transferred to the kernel. Please see the last 3 sentences of paragraph 7 (lines 144-151)  in Introduction and also the comment 3 as well as the extended paragraphs 3-6 in Section 5.

Round 2

Reviewer 1 Report

The authors have made careful revisions based on the review comments